# Environmental DNA metabarcoding reveals distinct fish assemblages supported by seagrass (*Zostera marina* and *Zostera pacifica*) beds in different geographic settings in Southern California

Tanner Waters[1,2]*, Zachary Gold[3], Adam Obaza[4], Richard F. Ambrose[1,5], Robert A. Eagle[1,2,6]*

1 Institute of the Environment and Sustainability, University of California, Los Angeles, CA, United States of America, 2 Center for Diverse Leadership in Science, University of California, Los Angeles, CA, United States of America, 3 NOAA Pacific Marine Environmental Laboratory, Seattle, WA, United States of America, 4 Paua Marine Research Group, Long Beach, CA, United States of America, 5 Department of Environmental Health Sciences, Jonathan and Karen Fielding School of Public Health, University of California, Los Angeles, CA, United States of America, 6 Atmospheric and Oceanic Sciences Department, University of California, Los Angeles, CA, United States of America

* tannerwaters@g.ucla.edu (TW); robeagle@g.ucla.edu (RAE)

## Abstract

Seagrass beds are disappearing at a record pace despite their known value to our oceans and coastal communities. Simultaneously, our coastlines are under the constant pressure of climate change which is impacting their chemical, physical and biological characteristics. It is thus pertinent to evaluate and record habitat use so we can understand how these different environments contribute to local biodiversity. This study evaluates the assemblages of fish found at five *Zostera* beds in Southern California using environmental DNA (eDNA) metabarcoding. eDNA is a powerful biodiversity monitoring tool that offers key advantages to conventional monitoring. Results from our eDNA study found 78 species of fish that inhabit these five beds around Southern California representing embayment, open coastal mainland and open coastal island settings. While each bed had the same average number of species found throughout the year, the composition of these fish assemblages was strongly site dependent. There were 35 fish that were found at both open coast and embayment seagrass beds, while embayment seagrass sites had 20 unique fish and open coast sites had 23 unique fish. These results demonstrate that seagrass fish assemblages are heterogenous based on their geographic positioning and that marine managers must take this into account for holistic conservation and restoration efforts.

## 1 Introduction

Seagrass ecosystems are ecologically, economically, and culturally significant in California. They provide dozens of ecosystem services including acting as juvenile fish nurseries [1],

**Data Availability Statement:** All analyses were conducted in R and made available here https://github.com/tannerwaters/SoCal_Seagrass.git.

**Funding:** TW is supported by the UCLA Cota-Robles Fellowship, the National Science Foundation Graduate Research Fellowship, the Switzer Foundation Fellowship, and the UCLA Center for Diverse Leadership in Science Fellowship. RE acknowledges support from the Anthony & Jeanne Pritzker Family Foundation (https://www.ajpff.org/) to UCLA IoES. This research was funded by a grant from the David and Lucile Packard Foundation no. 2020-70931 to the Center for Diverse Leadership in Science at UCLA (https://www.packard.org/) and a research award from the La Kretz Center for California Conservation Science and Stunt Ranch Santa Monica Mountains Reserve. The authors acknowledge Contribution Number 5506 from NOAA Pacific Marine Environmental Laboratory (PMEL). The funders had no role in study design, data collection and analysis, decision to publish, or preparation of the manuscript.

**Competing interests:** The authors have declared that no competing interests exist.

providing invertebrates habitat and food [2], protecting coastlines [3], stabilizing sediment [4], purifying water [5], sequestering carbon [6] and mitigating ocean acidification [7]. Seagrass beds are also known to be one of the most productive ecosystems on earth and hotspots for biodiversity in coastal systems [1,8]. They have been estimated to provide ecosystem services of over $19,000 per ha of meadow per year [9]. This places the worth of their ecosystem services higher than tropical forests, coral reefs, and mangroves per unit area [9]. However, over 29% of seagrass has disappeared globally since 1879 [10] and over 90% within certain parts of California [11]. The trend of seagrass loss has accelerated from 0.9%/year prior to 1940 to 7%/year since 1990 [10]. In order to fully evaluate the impact of this seagrass loss and provide justification for restoring these environments, it is necessary to evaluate the extent of the services they provide.

Seagrass beds are able to support high levels of biodiversity because they provide three-dimensional structure to an otherwise bare soft bottom seafloor. This vegetation provides a foundation for algae and epibionts to grow, which creates the basis of the ecosystem's food web. Associated seagrass species feed on the seagrass blades and associated epiphytes live on the blades. These species also use the seagrass' physical feature as protection from predators [12]. Fish diversity in particular is high within these habitats due to their dependence on seagrass as a nursery habitat [1]. Seagrass provides structural complexity for fish to attach their eggs to and for juvenile fish to hide from predators. While seagrass is known to increase survival rates compared to bare sand, seagrass is also shown to increase juvenile growth more than bare habitats and other structured habitats [13].

These beds are under constant stress of changing ocean conditions such as increased temperature, eutrophication, physical damage/removal and fishing pressures, Surveys done just ten years ago in the area may already be outdated in providing us with an understanding of the currently supported marine biodiversity [14–18]. Thus, there is a growing need to routinely monitor seagrass beds not just within embayments, but also on the open coast and Channel Islands within Southern California.

In order to assess the community composition of seagrass beds, this study employs environmental DNA (eDNA) metabarcoding. eDNA is the methodology of collecting free-floating DNA and cells that have been excreted or secreted from organisms [19]. This DNA is extracted and certain gene regions, known as barcodes, are amplified and sequenced to reveal species presence or absence for broad-scale biodiversity, predator diet analysis and trophic interactions [20]. eDNA retains some key advantages in biodiversity monitoring. Firstly, eDNA can differentiate between morphologically similar species [20]. This is especially important in seagrass beds that are used as nurseries where visual surveys may be unable to identify juveniles down to the species level [20]. Conventional surveys require taxonomic identification by an expert which could introduce errors from possible misidentification. Secondly, eDNA has been shown to better detect rare and cryptic species that are more easily overlooked in conventional methods including highly camouflaged and sediment inhabiting taxa that are difficult to detect using visual surveys [20]. Thirdly, eDNA sampling is logistically less complex in the field than visual surveys, which allows researchers to take a greater number of samples across broader spatial and temporal ranges [21]. Lastly, eDNA has often been demonstrated to be cheaper, more sensitive and able to detect more species when directly compared to traditional methods of biodiversity monitoring [22].

Environmental DNA has been shown to be a powerful tool when surveying seagrass habitats. Researchers have previously employed this method via water column collection [23–28] and sediment sampling [29,30]. A number of these seagrass studies have demonstrated that when directly compared to a conventional survey method, eDNA was able to detect a higher number of species [23–25]. Other studies emphasized the importance of using concurrent

eDNA and conventional survey techniques in revealing the full scope of biodiversity [26,27,30]. Despite the literature support of eDNA's use in seagrass monitoring, there has been no eDNA surveys done on fish communities of Southern California seagrass beds, which sit in a very specific biogeographic position of a productive upwelling region for both island and mainland populations. In Southern California, some of the seagrass population is within marine protected areas and others within heavily human impacted urban coastal environments.

Environmental DNA approaches do have known limitations that warrant consideration. First, it is important to note the influence of taxonomic assignment on data output and interpretation. The accuracy of taxonomic assignments are largely driven by two features: barcode choice and reference database completeness. For example, a commonly used barcode used for fish diversity globally, the MiFish Universal primer set, is unable to resolve the majority of fish in the *Sebastes* (Rockfish) genus [31], an environmentally and commercially important species in California. Thus, without the use of an additional barcode, the MiFish Universal primer set fails to resolve *Sebastes* species. While only a small number of rockfish are known to inhabit Southern California seagrass beds, it is still worth noting that their species level resolution is not possible utilizing this marker set alone. Likewise, accurate taxonomic assignment can only be achieved with comprehensive reference barcode databases that contain sequences for all monitored species [32]. Fortunately, extensive efforts have been made in the California Current Large Marine Ecosystem to sequence the vast majority of marine fishes [33].

In addition to taxonomic assignments, interpretation of eDNA metabarcoding data is influenced by detection probabilities. Detection probabilities are a function of both the total concentration of DNA in the environment and assay efficacy [34]. The total concentration of DNA in the environment is a function of shedding rates, degradation [35], and fate and transport in marine systems [36,37] while assay efficiency for a given taxa is a function of methodological choices including volume filtered, inhibition, and PCR driven amplification bias among many others [38,39]. Despite these limitations and biases, here we use well established marine eDNA assays with demonstrated efficacy in Southern California coastal marine ecosystems [40–44].

This study tests the utility of eDNA methods to provide seasonally resolved fish survey information in five *Zostera sp*. beds around Southern California with diverse biogeographic contexts like heavily human impacted embayments, open ocean coastal and island locations. Our aim is to better characterize the community composition of local seagrass beds as well as understand the benefits and limitations of using eDNA compared to conventional survey methods in coastal ecology biodiversity monitoring.

## 2 Materials and methods

### 2.1 Sample collection

We conducted our study of Southern California *Zostera* beds off the coast of Malibu, CA, Catalina Island, CA and Newport Bay, CA seasonally during 2019–2020. We collected these samples seasonally in Summer (July/August 2019), Fall (November 2019), Winter (February 2020) and Spring (May 2020). No permits were required for this work.

We sampled five *Zostera sp*. beds around Southern California: Amarillo, Two Harbors, Big Geiger Cove, Inner DeAnza Peninsula, and Outer DeAnza Peninsula. We collected additional samples at a sandy bottom control site on Catalina: Cherry Cove to compare with the two seagrass sites on Catalina- Big Geiger Cove and Two Harbors (Table 1).

These five sites represent distinct geographic locations of seagrass habitat within Southern California. These sites are grouped by their geography including open coast seagrass beds

**Table 1. Information of sampling design for the seagrass and control sites.**

| Site | Environment | Geographic Type | Depth | Coordinates | Location | Visual Survey |
|---|---|---|---|---|---|---|
| **Amarillo** | *Z. pacifica* | Open Coast- Mainland | 12.5 m | 34.02755, -118.700084 | Malibu, CA | Yes-Fall |
| **Two Harbors** | *Z. marina* | Open Coast- Island | 5.8 m | 33.443405, -118.49843 | Catalina Island, CA | Yes-Fall |
| **Big Geiger** | *Z. marina* | Open Coast- Island | 7.9 m | 33.459704, -118.517454 | Catalina Island, CA | Yes- Summer |
| **Inner DeAnza Peninsula** | *Z. marina* | Embayment | 2.42 m | 33.619506, -117.90291 | Newport Bay, CA | No |
| **Outer DeAnza Peninsula** | *Z. marina* | Embayment | 2.07 m | 33.619269, -117.901692 | Newport Bay, CA | No |
| **Cherry Cove** | Bare sand | NA | 7.2 m | 33.45129, -118.50195 | Catalina Island, CA | No |

(mainland- Amarillo, island- Big Geiger Cove and Two Harbors) and embayment seagrass beds (Inner DeAnza Peninsula and Outer DeAnza Peninsula) (Fig 1).

We employed the eDNA collection method of Curd et al. 2019 [45]. First, we collected seawater samples at depth directly above the seagrass beds using a 5L Niskin bottle. From the Niskin, we transferred one liter of seawater to a Kangaroo enteral feeding bag in triplicate. We immediately filtered the seawater through a sterile 0.22 μm Sterivex cartridge filter (Millipore-Sigma, Burlington, MA, USA) using a peristaltic pump until the 1L was fully through our cartridge. We capped the filters and stored them on dry ice during sampling until we returned to the lab where they were stored at -20°C in the lab. During each day of sampling, we filtered one liter of Milli-Q water through the same process for a negative field control [46].

## 2.2 DNA extraction and library preparation

We extracted our eDNA from the Sterivex cartridge using a modified DNeasy Blood & Tissue Kit protocol (Qiagen Inc., Germantown, MD) optimized for increased eDNA yield [47]. Library preparation followed a modified protocol [35]. Using PCR, we amplified the extracted DNA using the Mifish Universal Telost 12S primer set [48]. We used a 25 μL reaction composed of 12.5 μL QIAGEN Multiplex Taq PCR 2x Master Mix,1.5 μL of molecular grade H2O, 5 μL of forward primer (2 mM), 5 μL of reverse primer (2 mM), and 1μL of sample DNA. Our cycling conditions consisted of a touchdown PCR profile with an initial denaturation at 95°C for 15 min, followed by 13 cycles of 95°C for 30s, beginning annealing at 69.5°C for 30 seconds which decreases in temperature 1.5°C per cycle until 50°C, extension at 72°C for 1 minute. After the 13 cycles we did 24 cycles of 95°C for 30s followed by annealing at 50°C and an extension of 72°C for 1 minute. We use a final extension for 10 min at 72°C. A negative PCR control of molecular grade water was added following the same protocol. We verified amplification success by checking product size on a 2% agarose gel electrophoresis stained with SybrSafe.

After amplification, we modified the samples by adding individual Nextera unique dual indices (Illumina, San Diego, CA, USA). We used a 25 μL reaction composed of 12.5 μL Kapa Hifi MasterMix (Kapa Biosystems, Sigma Aldrich, St. Louis, MO, USA), 6.25 μL of molecular grade H2O, 1.25 μL of index and 5 μL of DNA from the PCR sample. Our PCR cycling parameters for indexing consisted of denaturation at 95°C for 5 min, followed by 8 cycles of denaturation at 98°C for 20s, annealing at 56°C for 30s, and extension at 72°C for 3 min, and then a

## Map of Sampling Sites

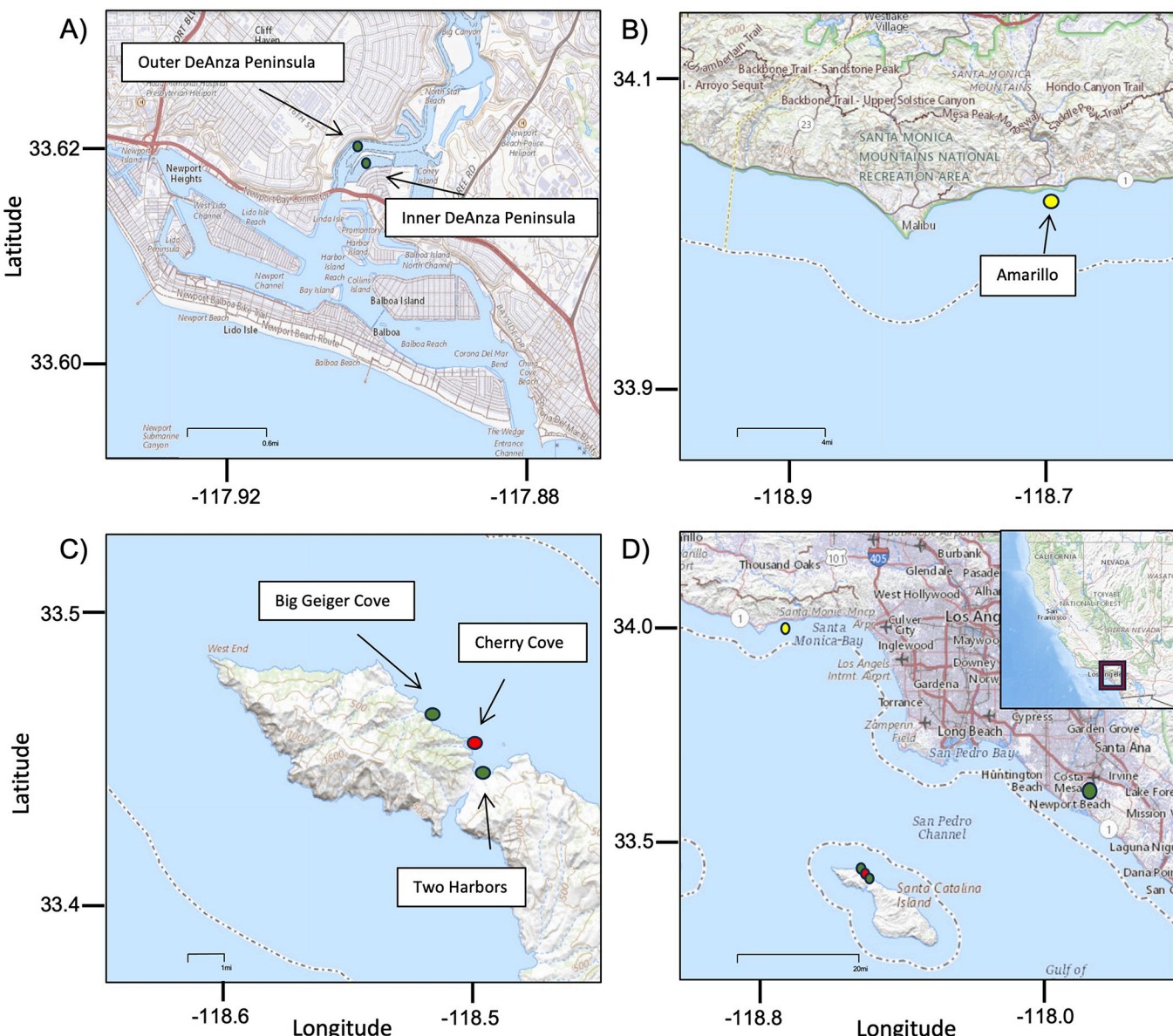

**Fig 1.** A) Map of embayment seagrass beds. B) Map of Open Coast Mainland seagrass beds. C) Map of Open Coast Island seagrass beds. D) Map of all sites. Yellow indicates *Zostera Pacifica*, green indicates *Zostera Marina*, and red indicates our no seagrass site. Maps from USGS National Map Viewer under a CC BY 4.0 license (2022): https://apps.nationalmap.gov/viewer/.

final extension at 72°C for 5 min. We verified amplification success by checking product size on a 2% agarose gel electrophoresis stained with SybrSafe.

We cleaned the resulting libraries using Omega BioTek Mag-Bind RXNPure Plus beads (Omega Bio-tek, Inc., Norcross, GA, United States). We then measured DNA concentration of each sample with the Qubit dsDNA Broad Range DNA Quantification Assay (Thermofisher Scientific, Waltham, MA, USA). Samples were then pooled in equal copy number. The final

library was sequenced at UCLA's Technology Center for Genomics and Bioinformatics (TCGB) on an Illumina NextSeq V2 PE 150 Cycles- Mid Output spiked with 12% PhiX.

## 2.3 Bioinformatics

We used the *Anacapa Toolkit* [45] for amplicon sequence variant parsing, taxonomic assignment, and quality control. The quality control step of the Anacapa Toolkit trims extraneous adapter sequences used to identify each unique sample, removes low quality reads, and sorts reads by metabarcode primer sequence. The amplicon sequence variant (ASV) parsing step uses DADA2 [49] to dereplicate our metabarcodes. Next the Anacapa toolkit module assigns taxonomy to ASVs using Bowtie 2 [50] and a Bowtie 2-specific Bayesian Least Common Ancestor (BLCA) algorithm [51].

For the fish primer set, taxonomic assignment was conducted following benchmarking by Gold et al. (2021) using a taxonomic cutoff score of 60 and minimum alignment of 80% [43]. Taxonomy was first assigned using the curated regional database of California Current Large Marine Ecosystem fishes to identify native taxa. We then re-assigned the taxonomy using the global *CRUX* generated database to identify non-native and non-fish species. Taxonomic assignments of ASVs were synonymized between both methods by prioritizing higher resolution assignments (i.e. species level vs. genus level).

We then implemented a decontamination procedure to eliminate poorly sequenced samples and remove potential sources of contamination [44,52–54]. Importantly, we applied a site occupancy modeling framework to retain only ASVs that occurred in high prevalence across locations and stations. For these analyses, we removed all non-fish taxa from the resulting data. All remaining ASV's had their read counts converted into the eDNA index [53]. The eDNA index transformation is conducted by first normalizing all reads for a particular sequence by the total number of reads in each sample, then scaling those proportions to the largest observed proportion for that sequence across all samples. This results in a sequence-specific (species-specific) scaling between 0 to 1, where 1 is the sample with the highest number of reads for a given species and 0 is the least.

## 2.4 eDNA data analysis

We tested if our sequencing depth reached species saturation for our samples using a rarefaction curve. In order to test if our eDNA field samples fully captured the species richness of the site, we used an iNext package [55] to model a site-specific species accumulation curve. We then ran a piecewise regression analysis to identify the breakpoint in the rate of species capture with the *R* package *segmented* [56]. Breakpoint analysis is the statistical method for showing the significant point in which the segmented regression changes slopes and thus where we begin to reach saturation for our sample's species discovery.

Next, we measured total species richness to compare alpha diversity between seagrass sites and sandy bottom and seasonally within seagrass sites. Total species richness was compared using a generalized linear model (GLM) with a Poisson regression and significant groups were determined using a tukey contrasts multiple comparisons of means test.

To test for differences in community composition (beta diversity), the eDNA indexes for the samples were converted into Bray-Curtis dissimilarity distances [52]. We tested for differences in community structure by site and season using an adonis PERMANOVA followed by a multivariate homogeneity of group dispersions test BETADISPER [57]. Community composition was visualized using non-metric multidimensional scaling (NMDS) [57]. Closer grouped data points indicate more closely related community composition in both species richness and diversity.

## 2.5 Visual fish surveys

We paired visual scuba fish surveys with our eDNA fish surveys. Visual surveys were taken at 1)Amarillo, 2) Big Geiger Cove and 3) Two Harbors. These three surveys occurred during the same month as our eDNA surveys but not on the same day. The timed roving visual surveys are described within Obaza et al., 2022 [18]. Briefly, we took six visual surveys at each site with three within the bed and three along the edge. We took each survey for 3–6 minutes each and recorded the fish species observed. We compared the presence or absence of species found by eDNA and visual surveys to identify strengths and limitations of both survey approaches.

## 3 Results

### 3.1 Species richness

The NextSeq generated over 10 million reads that passed quality control. Of these reads, 9.8 million reads representing 95 samples, 76 field samples and 19 blanks, passed the quality control of the Anacapa Toolkit. After taxonomic assignment we were left with 6.8 million reads representing 324 ASVs across 76 field samples. These reads represented 41 families, 69 genera, and 81 species of fish of which 40 families, 67 genera and 79 species were found within the seagrass sites (S1 Table in S1 Table). The ASV read counts were then converted into an eDNA index (S2 Table in S1 Table). Species are listed per site and per geographic type (S4 Table in S1 Table) as well as broken up seasonally by site (S5 Table in S1 Table). Sequences that could not be identified down to species are listed as *Genus sp.* and ASV's that blasted to more than one species are listed as *Genus species/species*.

Sample rarefaction curves showed that for each sample sequencing depth was sufficient to capture all species diversity within that collected sample (S1 Fig). Site-specific rarefaction curves modeled using the *iNext* package shows that at each site, the number of field replicates that were taken did not capture the full diversity of that site (S2 Fig). This analysis shows that for these sites roughly 12–19 samples were needed to reach the breakpoint in the rate of species diversity found per sample. (Amarillo: 14.1, Big Geiger Cove: 13.99, Cherry Cove: 12.39, Inner DeAnza Peninsula: 16.59, Outer DeAnza Peninsula: 19.01, Two Harbors: 15.22).

Comparisons between all sites found that the only significant difference in the mean number of species observed was between Outer DeAnza Peninsula and Amarillo (GLM $Pr(>|z|)$ = 0.03073) and Outer DeAnza Peninsula and Cherry Cove (GLM $Pr(>|z|)$ = 0.00183) (S6 Table in S1 Table, Fig 2). However seasonal variation in the number of species found at all combined seagrass sites was found to significantly differ with every season comparison except summer and fall (S6 Table in S1 Table, Fig 3). Species richness was highest during the spring and continued to decrease in the summer, fall and then winter.

### 3.2 Community composition

We performed NMDS in order to compare community structure. The NMDS shows that embayment, open coast, and island seagrass beds are compositionally distinct from one another while sites of similar geographies show significant overlap. NMDS ordination showed good clustering by both type (PERMANOVA p<0.001, $R^2$ = 0.34393, betadisper p>0.05) and season (PERMANOVA p<0.001, $R^2$ = 0.09577, betadisper p>0.05) (NMDS, Stress = 0.16, Fig 4).

Seagrass sites were grouped by their geographic location—Open Coast (Mainland and Island) and Embayment. There were 35 fish found at both geographic locations with 23 fish unique to open coast beds and 19 unique fish found at embayment beds (Table 2). This supports our NMDS clustering, which showed that geographically distinct beds have different

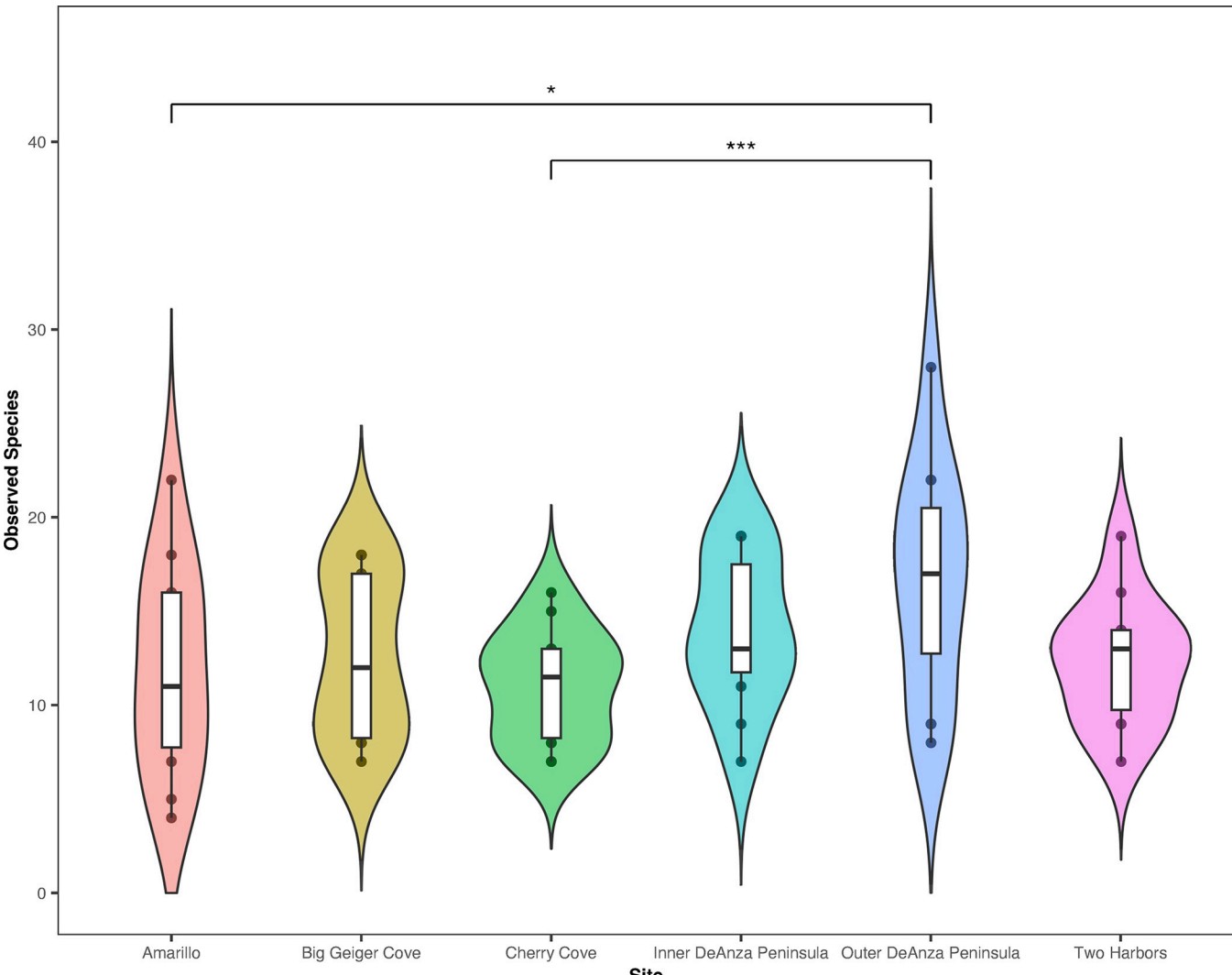

**Fig 2. Violin plot of species richness by seagrass site across all seasons.** A GLM and Tukey multiple comparison test shows the only significant difference between the number of species was found between Outer DeAnza Peninsula and Amarillo and Outer DeAnza Peninsula and Cherry Cove. * P ≤ 0.05, ** P ≤ 0.01, and *** P ≤ 0.001.

community composition than other types of beds. These beds remained distinct throughout the seasons.

### 3.3 Seagrass vs sandy bottom

Samples were taken at two Catalina seagrass beds and one nearby sandy bottom control site. Across the three sites, a total of 45 fish species were detected with eDNA. Of these 45 fish, 24 were shared between all three sites while 13 fish were only found in seagrass and 2 fish were only found at the sandy bottom control site (Fig 5; S7 Table in S1 Table).

### 3.4 eDNA vs visual species detections

For Big Geiger Cove, 8 species of fish were found by both methods during the summer time point. Environmental DNA detected an additional 7 unique species while scuba surveys found

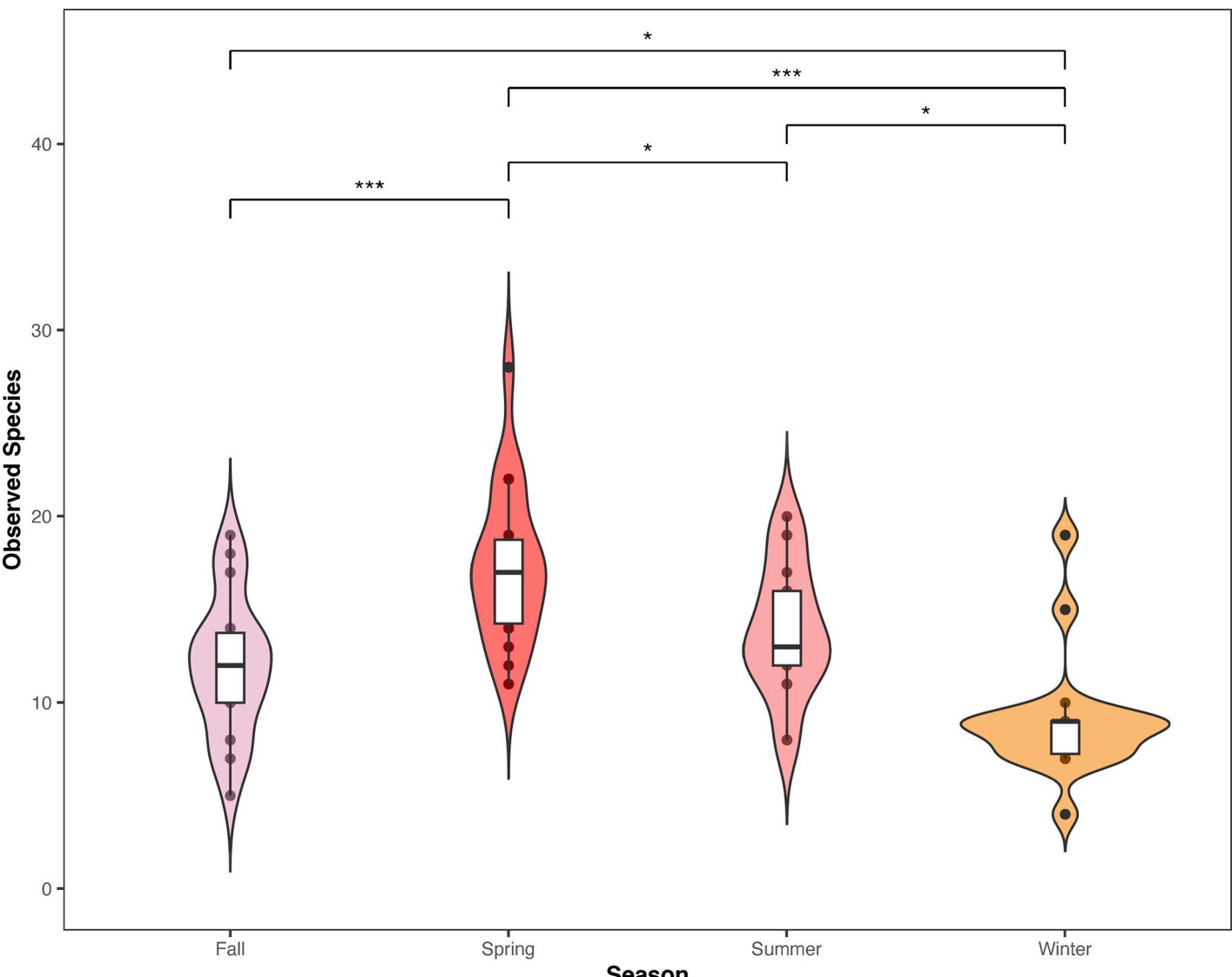

**Fig 3. Violin plot of species richness at all five seagrass sites by season.** The GLM shows there is a significant seasonal turnover in the number of species found at the beds throughout the seasons. * P ≤ 0.05, ** P ≤ 0.01, and *** P ≤ 0.001.

1 unique species (Fig 6; S4 Table in S1 Table). At Two Harbors during our fall time point, both methods captured 11 species of fish with eDNA detecting an additional 6 unique species and scuba surveys with 3 unique species (Fig 6; S8 Table in S1 Table). Amarillo showed the least congruence between survey methods. Both captured 2 similar species of fish but eDNA had 16 unique fish species and scuba surveys had 2 unique fish species (Fig 6; S8 Table in S1 Table).

## 4 Discussion

We successfully demonstrate the ability of eDNA to monitor fish assemblages in Southern California seagrass habitats. Environmental DNA was shown to capture a suit of taxa known to utilize these habitats based on previous surveys. We found distinct fish communities in embayment, mainland open coast and island open coast seagrass beds demonstrating the sensitivity of these approaches to characterize local biodiversity patterns. Environmental DNA was also

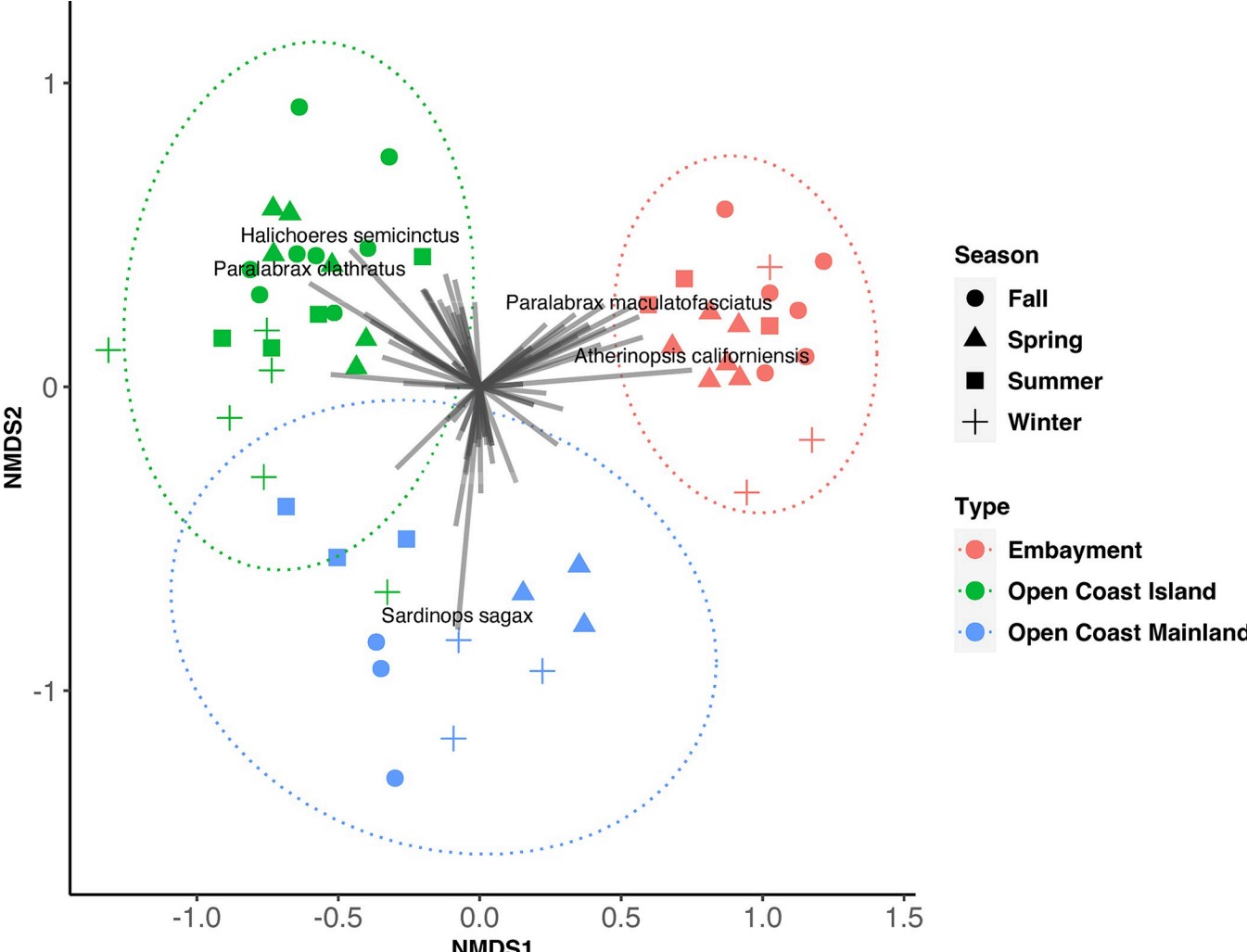

**Fig 4. NMDS visualization of Bray-Curtis similarities between the geographic seagrass types.** Types that are grouped closer to one another are more closely similar in both species richness and species count. NMDS shows that community composition of seagrass beds is more strongly dependent on their geographical location (embayment vs open coast mainland vs open coast island). Colors indicate type and shapes indicate season. Fish species were fit on the ordination where relative length indicates correlation between species and NMDS. The top five strongest associations are listed.

able to largely recapitulate visual surveys while detecting a broader array of marine fishes, demonstrating the efficacy of these approaches for future seagrass monitoring efforts.

## 4.1 eDNA captures biogeographic differences in fish assemblages

Our eDNA survey detected 78 unique species of fish within Southern California seagrass beds-48 species off the mainland, 48 species off the island and 54 species in the embayment. The number of fish surveyed is on par with or greater than other previous surveys in the area. From 1987 to 2010, embayment seagrass beds in San Diego Bay and Mission Bay were found to have supported 50 species of fish [14]. Newport Bay, the site of our embayment seagrass beds, has been surveyed since 2003; the latest monitoring survey published in 2020 found 26 species of fish [15]. One survey of open coast and island seagrass beds around the Northern Channel Islands and Santa Barbara coastline found that open coast beds supported 20 species of fish while island beds supported 41 species of fish [16]. In 2018, island seagrass beds along Catalina Island were recorded to support 28 species of fish [17,18].

**Table 2. Fish found at the different geographic seagrass beds.**

| Every Geographic Location | Open Coast Only[1] | Embayment Only |
|---|---|---|
| Barred sand bass | *Amphistichus sp./Hyperprosopon sp.*\* | Albacore |
| Bat eagle ray | Blind goby* | American shadow goby |
| Bay blenny | California lizardfish* | Bay goby |
| Bay pipefish | California skate* | Bocaccio rockfish |
| Black perch | California tonguefish* | California grunion |
| California halibut | Hornyhead turbot* | California killifish |
| California kingcroaker/Corbina | Pacific pompano* | Californian needlefish |
| California pilchard/Pacific Sardine | Pacific/Longfin sanddab* | Diamond stingray |
| California sheephead | Thornback guitarfish* | Diamond turbot |
| Californian anchovy | White croaker* | Eastern Pacific bonito |
| Californian salema | Bennett's flying fish^ | Gray smooth-hound |
| Chub mackerel | Blackeye goby^ | Longjaw mudsucker |
| Fantail flounder | Blacksmith^ | Shortfin weakfish |
| Flathead grey mullet | California scorpionfish^ | Slough anchovy |
| Garibaldi | *Cheilopogon sp.* ^ | Specklefin midshipman |
| Halfmoon | Horn shark^ | Spotted turbot |
| Haller's round ray | Largemouth blenny^ | *Thunnus sp.* |
| Jack silverside | Ocean whitefish^ | White/Queen croaker |
| Kelp bass | Opaleye^ | Yellowfin goby |
| Leopard shark | Spotted/Crevice/Striped kelpfish^ | |
| Mussel blenny | California clingfish | |
| Pacific barracuda | Giant kelpfish | |
| Pacific jack mackerel | Señorita | |
| Pacific sanddab | | |
| Reef finspot | | |
| Rock wrasse | | |
| *Sebastes sp.* | | |
| Shiner perch | | |
| Shovelnose guitarfish | | |
| Speckled sanddab | | |
| White seaperch | | |
| Xantic sargo | | |
| Yellowfin drum or croaker | | |
| Yellowtail amberjack | | |
| Zebra-perch sea chub | | |

[1]Species under 'Open Coast' with no symbol were found at both mainland beds and island beds, * represents species found only at open coast mainland beds and ^ represents species found only at open coast island beds. Sequences that could not be identified down to species are listed as *Genus sp.* and ASV's that blasted to more than one species are listed as *Species/Species*.

Our eDNA surveys detected distinct fish assemblages associated with open coast and embayment seagrass beds. The majority of fish species, 35, were shared between open coast and embayment beds and consisted of a mixture of rocky reef, soft bottom, and water column fish species. However, at geographically distinct sites, a noticeable pattern emerges.

For our embayment seagrass beds, 19 of its 54 fish species were only found at these two sites with the majority of them being soft bottom species. The embayment sites were the only sites to have fish associated with wetland species (California killifish and longjaw mudsucker)

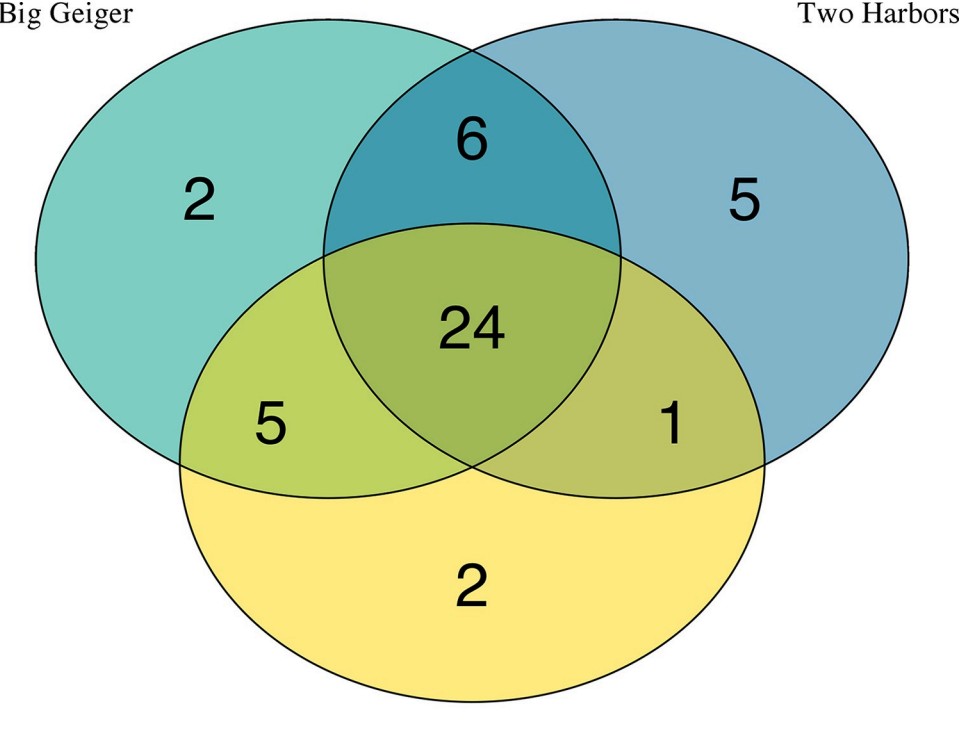

**Fig 5. Venn diagram of fish species detected by eDNA between two seagrass sites (Big Geiger Cove and Two Harbors) and one sandy bottom site (Cherry Cove).**

and also had the majority of detected estuary/bay associated species (Slough anchovy, diamond turbot, spotted turbot, American shadow goby, California needlefish, etc.). Notable at this site are two tuna species that were detected. To pick up signatures of its presence in a nearshore shallow environment in Southern California would be exceedingly rare. This is most likely an instance of fishers cleaning tuna catch in the back bay/harbor or an exogenous source of eDNA of this popular seafood in a highly urbanized area. On the other hand, our open coast seagrass beds had 23 unique species captured out of its 58 species total. In contrast to the embayment beds, the fish here were heavily linked to water column and rocky reef habitats. This distinction was further divided between beds located off the mainland and beds located off the island. Island sites had the highest proportion of what are typically rocky reef associated fish species (California scorpionfish, blacksmith, opaleye, blackeye goby, etc.) compared to open coast mainland beds, which had primarily soft bottom associated fishes. This difference in open coast vs embayment beds highlights the importance that other nearby coastal habitats play in the recruitment of fish to seagrass. Seagrass diversity and recruitment has been previously shown to be affected by distance from dispersal site [58], proximity to other habitats [59], and wind patterns [60]. Additional influences on fish diversity seagrass sites that may contribute to these differences beyond geographic setting of the meadow includes heterogeneity of environments surrounding the meadow [61], proximity to other seagrass sites [62,63], seagrass canopy height [64], and seagrass cover [65]. Although our in-situ design can't account for all possible influences on seagrass diversity, our surveys suggest that geographic location can impact up to half the found species at a given seagrass bed.

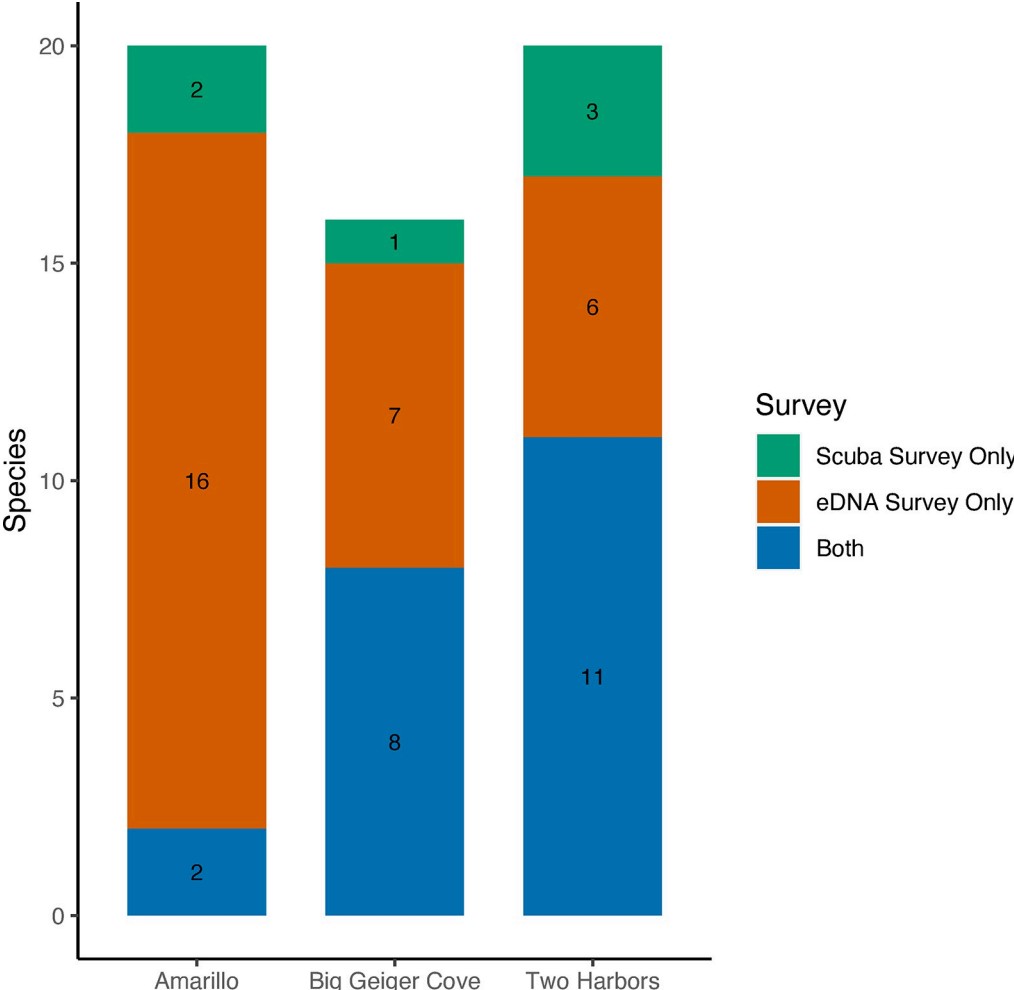

**Fig 6. The number of species observed by eDNA and conventional methods.** "Both" indicates species that were detected by both scuba surveys and eDNA surveys. "Scuba survey only" and "eDNA survey only" show the number of species that were uniquely detected by that method. Total number of species detected by scuba survey is "both" plus "scuba survey only" and total number of species detected by eDNA surveys is "both" plus "eDNA survey only".

One specific group of interest that showed geographic differences was elasmobranchs. Worldwide, shark and ray populations have decreased 71% since 1970 [66]. Sharks assert a top-down control on their ecosystem through their predation of lower-level taxa which has a direct impact on the success of seagrass meadow [67,68]. Elasmobranchs were well represented in the five beds surveyed, detecting one skate species, five ray species and three shark species. Similar to other fish in this survey, their habitat use was geographically varied. The bat ray, Haller's round ray, shovelnose guitarfish, and leopard shark were found at both open coast and embayment beds. The California skate, thornback guitarfish, and horn shark were only found in open coast sites while the diamond stingray and gray smoothhound were only found in embayment beds. Our results demonstrate the value of seagrass habitats to sharks and rays, encouraging continued conservation of this key marine habitat.

The differences in community composition of our seagrass sites emphasize the heterogeneity of seagrass associated fish assemblages which have been shown previously to be different between *Zostera* species [18] and now shown to be distinct across biogeographic regions. The

significant differences of fish assemblages in Southern California underscores the importance of protecting multiple seagrass habitats across the region. Currently, only 5.2% of the eelgrass in Southern California is protected in a marine protected area [69]. Our results strongly suggest that California's Ocean Protection Council and Department of Fish and Wildlife management efforts should consider both the quantity and biogeographic distribution of seagrass habitats in order to protect the greatest number of fish species.

## 4.2 Value of Southern California seagrass

California's oceans are an important part of the state's economy, bringing in a gross state product of $84 billion dollars per year and supporting over 1 million jobs [70]. Nearly one quarter of the gross state product and jobs come out of Los Angeles County alone [70]. Marine vegetation, such as seagrass in Los Angeles, directly impacts the output of our oceans. Our eDNA recorded both commercially and recreationally important fish to California. Commercially, this includes California halibut which was the 7th largest commercial fishery in 2022 totaling 992,021 pounds valued at $5.4 million [71]. Recreationally, seagrass was home to the 5th (Flatfish, e.g. California halibut and Pacific sanddab), 8th (California scorpionfish), 9th (Sea bass) and 10th (Ocean whitefish) most caught fish categories by pounds in 2022 [71]. Seagrass meadows support these economically important species by acting as both nurseries for juvenile fish as well as habitat and food for adult fish. Four of these species was found at all seagrass bed types (California halibut, Pacific sanddab, kelp bass and barred sand bass) while the others were found in geographically distinct beds (ocean whitefish and California scorpionfish, open coast -island). Our results demonstrate the value of eDNA approaches for monitoring commercially important fish species and their utilization of key seagrass habitats, providing further evidence for the efficacy of eDNA approaches for routine marine biodiversity monitoring efforts.

## 4.3 Seasonality of seagrass fish assemblages

Average seasonality for the sites followed a general pattern with the highest number of species being found in the spring followed by summer/fall and the least in winter. This follows conventional patterns of fish breeding in *Zostera* beds in the late spring to early summer periods which would increase the diversity present [72]. There were two sites that had notable exceptions to this. The first was Big Geiger Cove off Catalina Island, which saw the greatest number of fish species in the fall. Tanner et al., 2019 found that coastal seagrass use off Catalina attracted young of year kelp bass around the fall months with a significant amount of biomass being exported to other coastal habitats in the winter months [73]. Preferential nursery use based on geographic location could account for differences in number of species recorded compared to the other sites. The second site was Inner DeAnza Peninsula, which found the highest number of species in winter and subsequently decreased until fall, although it remained relatively constant during the year with a roughly one species difference per season. Inner deAnza's seagrass bed, being high within Newport Bay and protected by a sand bank, could provide a steady and safe environment for the fish in the embayment that the other sites could not.

## 4.4 Seagrass boosts higher diversity over sandy bottom site

The loss of seagrass has been shown to cause the rapid shift and subsequent decline in species richness in those areas experiencing decline [74–76]. In California, loss of seagrass has been linked to decreased epifaunal diversity [74] and shifts in fish assemblages [77]. This study aimed to evaluate the effect seagrass has on community composition by surveying three sites

off Catalina Island- two of which were seagrass meadows found in Big Geiger Cove and Two Harbors, and one adjacent sandy bottom cove.

Between the three sites, there was substantial overlap in the majority of species found. Twenty-four species were found at both the seagrass sites and the sandy bottom site. eDNA found that there was higher diversity at the seagrass sites with 13 unique species captured and only 2 unique species found at the sandy bottom site. While NMDS shows that there is overlap between the sites, distinct communities were grouped together. Cherry cove was most similar to Two Harbors, a site of fragmented seagrass patches, in terms of community composition throughout the seasons. The sandy bottom site was even more dissimilar to Big Geiger, which is a cove with a continuous patch of seagrass. This suggests that seagrass beds density and size may play a part in their role as fish habitat. While it is unsurprising that a number of fish were found between both seagrass and sandy bottom coves, due to daily movement of fish in the ocean, seagrass is still important to these overlapped species as they rely on it for food and habitat. The species found only within the seagrass off Catalina included species known to use seagrass as nurseries (leopard shark and shiner perch), foraging grounds (shovelnose guitarfish), and habitat (Californian salema, bay blenny, and barred sand bass) while the two species found only at the sandy bottom site were common coastal pelagics (mackerel tuna) and known to hide under sand to attack pray (Pacific angelshark) [78].

One particular species of interest that was found in both seagrass meadows but not in the sandy site was the largemouth blenny, *Labrisomus xanti*. The largemouth blenny is a species native to Mexico with its previous range extending to the coast of Baja California [79]. The years of 2013–2015 brought an unusually warm ENSO event which caused a larger than normal distribution of warm water within the Pacific. Due to this, the first sighting of the largemouth blenny outside of its historical range was in La Jolla, California and Catalina Island in 2015 [79]. A recent study by Stockton et al., 2021 evaluating their population off of Catalina Island found this species to be positively correlated with rocky habitat and negatively correlated with sandy habitats [80]. Their preference for structured habitats, along with known associations of other blenny species with seagrass, could point to seagrass playing a role in the future expansion of fish ranges with climate change.

## 4.5 Comparison of eDNA and visual fish sampling method

Previous literature has shown that environmental DNA often captures a larger number of species when compared directly to conventional methodologies [22]. This has been shown to be true for surf zone fish communities in Southern California [81]. The result of our comparison is concurrent with these previous findings by showing that eDNA captured the majority of fish the conventional method did and found a greater number of additional fish species that the conventional method wasn't able to do.

At Big Geiger Cove, eDNA captured 8 of the 9 (88.8%) species that scuba surveys captured plus 7 additional species. At Two Harbors, eDNA captured 11 of the 14 (78.5%) species that scuba surveys captured plus 6 additional species. The known habitat preference of the majority of fish eDNA captured support the conclusion that these are likely true positives. One possible reason for the discrepancy between survey methods was that they were taken within the same month but not at the same time, so the fish could have truly not been there during the other survey methods. Other possible explanations for being missed in the visual survey is that some species attach their eggs to seagrass (jack silverside), use seagrass at night (ocean whitefish and California scorpionfish) or engage in camouflage (fantail flounder and California flounder), which would make it harder for visual surveys to observe them. The fish that were exclusively found in the visual surveys were only counted 1–2 times, which suggests eDNA surveys may

have lower probabilities of detection for rarer taxa with presumably lower total DNA in the environment. These results align well with previous work comparing eDNA and manual methods [81–83].

When looking at the comparison between the two methods at Amarillo, eDNA captured 2 of the 4 (50%) species that scuba surveys captured plus 16 additional species. This example highlights eDNA as being less dependent on ambient conditions during sampling. Turbidity, low light, minimal visibility, and rough ocean conditions can all impact a scuba divers ability to see and properly identify fish species. Since eDNA relies on capturing DNA in the water column, these issues do not impact a researcher's ability to properly survey an area. While visual surveys provide additional information that eDNA surveys cannot, such as abundance or fish length, environmental DNA was able to detect a higher number of species at these three sites. By relying on solely conventional methods, environmental managers could possibly miss rare or ecologically and economically important fish species which could alter how they structure their conservation efforts. The use of eDNA is important for characterizing the full extent of a habitat's biodiversity.

## 4.6 Benefits and drawbacks of eDNA

Environmental DNA is known to provide a number of benefits including differentiating morphologically similar species [20], detecting cryptic species [84], capturing a greater number of species compared to conventional methods [22], and being able to sample at greater spatial and temporal scales due to ease of use [20]. Our eDNA survey of seagrass beds around Southern California was able to confirm these benefits. The study's sampling regime of five beds across Southern California took only 3 days per season due to the relative ease of eDNA field sampling. The survey results were able to differentiate between the juvenile fish species, that use seagrass as a nursery, which often look morphologically similar. Within this study, eDNA was also able to capture a number of rare and cryptic species. One rare species found within these seagrass meadows is the vulnerable diamond stingray (*Hypanus dipterurus*), which is of management concern due to its International Union for Conservation of Nature (IUCN) status. Environmental DNA was also able to detect cryptic species including those which might avoid conventional detection through camouflage (Pacific sanddab, speckled sanddab, California halibut, diamond turbot, and bay pipefish), burial (blind goby), and through their small size (California clingfish, American shadow goby, muscle blenny). By capturing free-floating DNA in the water column, researchers can circumvent some of the obstacles that visual identification has.

Despite existing literature supporting the use of eDNA for the surveying of marine ecosystems, there are limitations. One such limitation is the identification of false positives, i.e. fish that were detected in our eDNA sample without actually being in the seagrass. Fish that were not necessarily occupying the seagrass could have their DNA transported into the bed and captured by our surveys. Previous work has consistently demonstrated that within coastal marine ecosystems, fate and transport are less of a concern as marine eDNA signatures tend to vary at a scale of ~50-800m with the higher end of this range being in the Puget Sound which has a much higher tidal transport than Southern California [40,43,44,85–88]. This range overlaps with the majority of the seagrass bed cover which ranges from 3,500m$^2$-31,000m$^2$ [17,89] so sampling in the center of the bed should reduce outside DNA input. Additionally, marine vegetation is known to slow hydrodynamic flow of water currents [90] which hypothetically shrinks the potential for DNA to be moved in or out of the system. The second limitation is not capturing the full species richness of the sites in our samples. Our analysis indicated that our sites needed roughly 12–19 samples to reach the breakpoint in the rate of species diversity

found per sample and that our sampling would benefit from an additional 1–7 extra samples taken over the course of the entire year. This is in line with other Southern California eDNA studies which found similar values for their sampling to reach saturation of marine fish biodiversity [41,43]. A third limitation of eDNA is that, in its current state, it is an assessment tool of species richness which limits our understanding of the data and its ecosystem function that may otherwise be understood from additional data taken from conventional surveys including size frequency, sex ratio, and absolute abundance data. Despite this, the information from eDNA still provides a valuable insight into local biodiversity.

## 5 Conclusion

Seagrass ecosystems are crucial habitats for fish within Southern California. Over 78 fish were documented through metabarcoding in the seagrass beds around Southern California. Community composition was found to be spatially and seasonally distinct with different geographic locations and seasons impacting which fish were found to utilize the seagrass. Our results of the environmental DNA methodology supported its use as a biodiversity monitoring tool for coastal ecosystems as it was able to provide additional information in detecting species that visual surveys did not. Visual survey and eDNA may yet be best employed as complementary approaches with visual methods providing information on other parameters such as fish length and encounter rate.

## Supporting information

**S1 Fig. Species richness sequence depth rarefaction.**
(TIF)

**S2 Fig. Site-specific rarefaction curves.**
(TIF)

**S3 Fig. NMDS visualization of Bray-Curtis similarities between seagrass sites.** NMDS shows that community composition of seagrass beds is more strongly dependent on their geographical location, i.e. in an embayment (Inner and Outer Newport), open coast (Amarillo) or island (Big Geiger and Two Harbors), than the season of sampling. Colors indicate site and shapes indicate season.
(TIF)

**S1 Table. Supplemental Tables 1–8.**
(XLSX)

## Acknowledgments

The authors acknowledge the Barber Lab at UCLA for access to laboratory facilities and to Dr. Jeroen Molemaker and the UCLA Zodiac crew for chartering our sampling.

## Author Contributions

**Conceptualization:** Tanner Waters, Robert A. Eagle.

**Formal analysis:** Tanner Waters, Zachary Gold.

**Funding acquisition:** Tanner Waters, Robert A. Eagle.

**Investigation:** Tanner Waters, Adam Obaza.

**Methodology:** Zachary Gold.

**Supervision:** Richard F. Ambrose, Robert A. Eagle.

**Writing – original draft:** Tanner Waters.

**Writing – review & editing:** Tanner Waters, Zachary Gold, Adam Obaza, Richard F. Ambrose, Robert A. Eagle.

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
