## [Decision Letter · Decision Letter 0]

4 Jul 2023

PONE-D-23-14325Environmental DNA metabarcoding reveals distinct fish assemblages supported by seagrass (Zostera marina and Zostera pacifica) beds in different geographic settings in Southern CaliforniaPLOS ONE

Dear Dr. Waters,

Thank you for submitting your manuscript to PLOS ONE. After careful consideration, we feel that it has merit but does not fully meet PLOS ONE’s publication criteria as it currently stands. Therefore, we invite you to submit a revised version of the manuscript that addresses the points raised during the review process.

We look forward to receiving your revised manuscript.

Kind regards,

Silvia Mazzuca

Academic Editor

PLOS ONE

Journal Requirements:

Additional Editor Comments:

Dear authors,

three independent reviewers took charge the revision of your manuscript. All of them agreed on the quality of the work done, but two required major revisions, the last one only minor. I kindly ask you to improve the manuscript following all the suggestions mad by the three reviewers and submit soon the revisied version.

best wishes

Silvia Mazzuca

Handling editor

Reviewers' comments:

Reviewer's Responses to Questions

**Comments to the Author**

1. Is the manuscript technically sound, and do the data support the conclusions?

Reviewer #1: Partly

Reviewer #2: Yes

Reviewer #3: Yes

2. Has the statistical analysis been performed appropriately and rigorously? 

Reviewer #1: Yes

Reviewer #2: Yes

Reviewer #3: Yes

3. Have the authors made all data underlying the findings in their manuscript fully available?

Reviewer #1: Yes

Reviewer #2: Yes

Reviewer #3: Yes

4. Is the manuscript presented in an intelligible fashion and written in standard English?

Reviewer #1: Yes

Reviewer #2: Yes

Reviewer #3: Yes

5. Review Comments to the Author

Reviewer #1: Environmental DNA metabarcoding reveals distinct fish assemblages supported by seagrass (Zostera marina and Zostera pacifica) beds in different geographic settings in Southern California

In the abstract, you mention the importance of temporal changes over time, but only reflect spatial changes. Please, add some information on how changes occurred over time.

The introduction starts off well, but then it includes text more appropriate for methodology and discussion sections. Also, it is missing a discussion on eDNA studies on fisheries or marine ecology. This is critical to allow the reader to understand the importance of the work presented in the manuscript.

Methods are fine. I added a few comments throughout. I’d suggest doing an NMDS using ‘open’ vs ‘embayment’, instead of just using sites. This might give you a tighter fit (lower stress).

Results have a few comments I added.

Discussion needs a quick mentioning on what the consequences management/conservation would be in using only visual surveys. Use existing studies that employ visual surveys and make conjectures on how their findings/conclusions would be different. This will add to the justification of using eDNA (probably). Also, in the abstract you mention upfront the importance of studies on diversity focusing on the temporal dimension, but you never collect temporal data. Seasonal data is not temporal in a multi-year sense. Either delete anything you mention about time or change it to seasons.

You did a great job. A few tweaks and it will be good for prime time.

Reviewer #2: This is an interesting paper looking at diversity in seagrass in embayment areas, mainland open coasts, and island open coasts, and identified distinct fish communities in each location using environmental DNA (eDNA). The findings emphasize the effectiveness of eDNA analysis in characterizing local biodiversity patterns. Additionally, the study demonstrated that eDNA analysis can closely replicate visual surveys while also detecting a wider range of marine fish species. This research is significant because it compares eDNA results with in-situ surveys, which is not commonly done in the literature and highlights the importance of conducting surveys in the exact sampling areas and approximately the same time as eDNA collection.

Therefore, this study is very important and very welcome. However, some major concerns prevent me from recommending the publication of this manuscript in its current form. Below, I have specified some major and minor issues which the authors could take into consideration to improve the overall quality of the manuscript.

The introduction, discussion, and literature cited of the study primarily focus on California, which overlooks the need for a broader context and comprehensive information on previous studies conducted on environmental DNA (eDNA) in seagrass meadows. These studies are highly relevant to the current research and should be adequately incorporated into the introduction to provide a more comprehensive understanding of the subject. Some relevant studies:

Angeles, I. B., Romero-Martínez, M. L., Cavaliere, M., Varrella, S., Francescangeli, F., Piredda, R., ... & Frontalini, F. (2023). Encapsulated in sediments: eDNA deciphers the ecosystem history of one of the most polluted European marine sites. Environment International, 172, 107738.

Qiu, S., Ooi, J. L. S., Chen, W., Poong, S. W., Zhang, H., He, W., ... & Loh, K. H.

(2023). Heterogeneity of Fish Taxonomic and Functional Diversity Evaluated by eDNA and Gillnet along a Mangrove–Seagrass–Coral Reef Continuum. Animals, 13(11), 1777.

Wesselmann, M., Geraldi, N. R., Marbà, N., Hendriks, I. E., Díaz-Rúa, R., & Duarte, C. M. (2022). eDNA Reveals the Associated Metazoan Diversity of Mediterranean Seagrass Sediments. Diversity, 14(7), 549.

Stat, M., John, J., DiBattista, J. D., Newman, S. J., Bunce, M., & Harvey, E. S. (2019). Combined use of eDNA metabarcoding and video surveillance for the assessment of fish biodiversity. Conservation Biology, 33(1), 196-205.

Introduction: The current organization of the introduction requires significant reorganization, starting with a general context to specific research gaps. Additionally, there is a notable absence of relevant literature that should be included to provide a comprehensive understanding of the topic.

Line 60 – 63: Explain before how is the structural complexity of seagrass meadows (three dimensional habitat – from belowground components with network of roots and rhizomes to aboveground components) and which components are important for fish. Also, not sure what do you mean with unstructured habitats, be more specific. Do you mean bare sand? Other macrophytes?

Line 83-88: The paragraph is better placed in the earlier sections where the significance of seagrass meadows is discussed: they are are hotspots for biodiversity but have high regression rates and therefore it is necessary to monitor seagrass meadows continuously due to the changing ocean conditions and their already current regression rates.

Line 94: edna is not only used to follow biodiversity also to understand the diet of animals, trophic interactions…etc. include citation

Line 97: include citation

Line 104: Also include that visual census surveys are limited in scope as they tend to focus on specific taxa and are conducted at a local scale. Additionally, these surveys require taxonomic experts to accurately identify and classify organisms. Therefore, there is a potential for misidentification, which can introduce errors or uncertainties in the collected data.

Line 106: Include in this section what has been done in edna in seagrass meadows (both in water and sediments) in the world and in California.

Methods: To understand better the sampling design, Include a table with date of collection, site, location, seagrass meadow/bare sand, open coast/embayment, depth, geographic coordenates, visual survey conducted or not

Lien 143-149: What were the depths of the sampling sites? Same depth among sites?

Line 164: use citation number to be conssitent though the text

Line 403: this paragraph fits better at the end of the discussion

Figure 1: Heading missing

Figure 6: how can be that the number of species found in both surveys is bigger than the sum of edna and scuba survey?

Figure 4: Fit fish species on the ordination so that their relative length indicate the correlation between species and the NMDS.

Reviewer #3: PONE-D-23-14325

This study used eDNA metabarcoding to examine fish species richness in 5 seagrass meadows in Southern California using one genetic marker. The paper is well written, has solid approach for the bioinformatics and statistical analyses. The results are set in the regional literature and provides a nice contribution to the growing literature on the utility of eDNA metabarcoding as a non-invasive method for assessing species richness in sensitive habitats.

I have suggestions for revisions to improve the scope and relevance of the study beyond California. Plus some general comments on other environmental characteristics that may be correlated with the differences between embayment and open coastal seagrass meadows that I think should be included in the literature review. No new analyses are necessary, as the sample size is too small (5 meadows) but I think a few key references on how the surrounding seascape and meadow characteristics may influence fish diversity within the sampled seagrass is needed.

I have provided some line edits below:

Line 69-81 – Although locally relevant, this paragraph is too detailed for a study introduction. It seems more suited for the discussion where the authors can report their results in the context of other studies. A literature review for this paragraph potentially more relevant would be to focus on how seagrass meadow characteristics and the surrounding seascape may influence biodiversity within the meadow. For example, the heterogeneity of environment outside the meadow, the size of the meadow, shoot density, temperature, salinity (e.g., Proudfoot et al 2023 https://doi.org/10.1007/s12237-023-01203-z, Chalifour et al 2019 https://doi.org/10.3354/meps13064, Gilby et al 2018 DOI: https://doi.org/10.3354/meps12394, and many more)

Line 375 – Some of relevant literature above could also be brought into the interpretation of results and acknowledge limitations of study, as multiple unmeasured factors likely influence observed species differences between sites.

Line 524 – the authors refer to eDNA metabarcoding as a useful a biodiversity monitoring tool for seagrass meadows, but in this paragraph on limitations, they fail to mention that eDNA metabarcoding is currently more of an assessment tool, or a snapshot of species richness. It can be examined over time but species richness is limited in its value for understanding ecosystem function. Just an acknowledgement that eDNA metabarcoding for biodiversity monitoring currently limited to richness (and at most rank abundance of dominant species – Skelton et al 2022 https://doi.org/10.1002/edn3.355

Figure 1 – inset map needed. Readers outside of Southern California need to be oriented to study site. Scale also needed (distance between sites could also be reported in results)

Not sure why figure captions are scattered throughout document rather at the end just before the figures. This made it difficult to find the correct caption.

Figure 2 – Include “Violin plot” in figure caption to ensure caption is descriptive of plot.

6. PLOS authors have the option to publish the peer review history of their article (what does this mean?). If published, this will include your full peer review and any attached files.

Reviewer #1: **Yes: **Ralf Riedel

Reviewer #2: No

Reviewer #3: No

---

## [Author Response · Author response to Decision Letter 0]

15 Aug 2023

Editor comments:

- We have updated the file naming.

In your Methods section, please provide additional information regarding the permits you obtained for the work.

- We have updated that no permits were required.

We note that Figure 1 in your submission contain [map/satellite] images which may be copyrighted.

-We have changed the map to use USGS National Map Viewer.

Please include captions for your Supporting Information files at the end of your manuscript, and update any in-text citations to match accordingly. 

-We have added captions for the supporting information.

Point by point responses to reviewers’ comments- Waters et al.

Reviewer #1: Environmental DNA metabarcoding reveals distinct fish assemblages supported by seagrass (Zostera marina and Zostera pacifica) beds in different geographic settings in Southern California

1. In the abstract, you mention the importance of temporal changes over time, but only reflect spatial changes. Please, add some information on how changes occurred over time.

Line 27: We have updated the abstract to reflect spatial changes rather than temporal changes. The sentence now reads “so we can understand how these different environments contribute to local biodiversity.” 

2. The introduction starts off well, but then it includes text more appropriate for methodology and discussion sections. Also, it is missing a discussion on eDNA studies on fisheries or marine ecology. This is critical to allow the reader to understand the importance of the work presented in the manuscript.

Line 91-101: We have added a paragraph on eDNA studies on seagrass ecology. The paragraph now reads “Environmental DNA has been shown to be a powerful tool when surveying seagrass habitats. Researchers have previously employed this method via water column collection [23-28] and sediment sampling [29-30]. A number of these seagrass studies have demonstrated that when directly compared to a conventional survey method, eDNA was able to detect a higher number of species [23-25]. Other studies emphasized the importance of using concurrent eDNA and conventional survey techniques in revealing the full scope of biodiversity [26, 27, 30]. Despite the literature support of eDNA’s use in seagrass monitoring, there has been no eDNA surveys done on fish communities of Southern California seagrass beds, which sit in a very specific biogeographic position of a productive upwelling region for both island and mainland populations. In Southern California, some of the seagrass population is within marine protected areas and others within heavily human impacted urban coastal environments.”

3. Methods are fine. I added a few comments throughout. I’d suggest doing an NMDS using ‘open’ vs ‘embayment’, instead of just using sites. This might give you a tighter fit (lower stress).

Thank you for the feedback in the methods. We confirmed that PLOS one accepts methods written in the first person. 

Lines 234-237 & 276-282: We have rerun the species richness comparison within sites and seasons using a generalized linear model with a Poisson link function and a tukey contrasts multiple comparisons of means test for significance which is more appropriate for this count data. The updated results have been replaced and the difference between sites and seasons is largely the same. The changes in methods occur on lines 234-237 which now reads “Next, we measured total species richness to compare alpha diversity between seagrass sites and sandy bottom and seasonally within sites. Total species richness was compared using a generalized linear model (GLM) with a Poisson regression and significant groups were determined using a tukey contrasts multiple comparisons of means test.” The changes also occur in the results in lines 276-282. 

Line 293-297: We have added the NMDS plot of sites with an NMDS of geographies on line 293-297. The NMDS of geographic type more clearly shows the main findings of the paper although stress remains unchanged as we did not change any of the underlying data but regrouped and colored by a different variable.

4. Results have a few comments I added.

Line 276-282, 284-290 & 293-297: We have updated the results of the alpha and beta tests from the updated methods mentioned previously. 

5. Discussion needs a quick mentioning on what the consequences management/conservation would be in using only visual surveys. Use existing studies that employ visual surveys and make conjectures on how their findings/conclusions would be different. This will add to the justification of using eDNA (probably). 

Line 517-521: We have added how the use of just visual surveys might affect conservation with the following “By relying on solely conventional methods, environmental managers could possibly miss rare or ecologically and economically important fish species which could alter how they structure their conservation efforts. The use of eDNA is important for characterizing the full extent of a habitat's biodiversity.”

6. Also, in the abstract you mention upfront the importance of studies on diversity focusing on the temporal dimension, but you never collect temporal data. Seasonal data is not temporal in a multi-year sense. Either delete anything you mention about time or change it to seasons.

Line 27: We have updated the use of temporal to reflect the seasonal data that was collected.

7. You did a great job. A few tweaks and it will be good for prime time.

Thank you for the kind words and for the time and responses you put into this review. 

Reviewer #2: This is an interesting paper looking at diversity in seagrass in embayment areas, mainland open coasts, and island open coasts, and identified distinct fish communities in each location using environmental DNA (eDNA). The findings emphasize the effectiveness of eDNA analysis in characterizing local biodiversity patterns. Additionally, the study demonstrated that eDNA analysis can closely replicate visual surveys while also detecting a wider range of marine fish species. This research is significant because it compares eDNA results with in-situ surveys, which is not commonly done in the literature and highlights the importance of conducting surveys in the exact sampling areas and approximately the same time as eDNA collection.

Therefore, this study is very important and very welcome. However, some major concerns prevent me from recommending the publication of this manuscript in its current form. Below, I have specified some major and minor issues which the authors could take into consideration to improve the overall quality of the manuscript.

Thank you for your comments of our study being important and welcome. We appreciate the reviewer’s suggestions and have incorporated them into the revisions of the manuscript. 

8. The introduction, discussion, and literature cited of the study primarily focus on California, which overlooks the need for a broader context and comprehensive information on previous studies conducted on environmental DNA (eDNA) in seagrass meadows. These studies are highly relevant to the current research and should be adequately incorporated into the introduction to provide a more comprehensive understanding of the subject. Some relevant studies:

Angeles, I. B., Romero-Martínez, M. L., Cavaliere, M., Varrella, S., Francescangeli, F., Piredda, R., ... & Frontalini, F. (2023). Encapsulated in sediments: eDNA deciphers the ecosystem history of one of the most polluted European marine sites. Environment International, 172, 107738.

Qiu, S., Ooi, J. L. S., Chen, W., Poong, S. W., Zhang, H., He, W., ... & Loh, K. H.

(2023). Heterogeneity of Fish Taxonomic and Functional Diversity Evaluated by eDNA and Gillnet along a Mangrove–Seagrass–Coral Reef Continuum. Animals, 13(11), 1777.

Wesselmann, M., Geraldi, N. R., Marbà, N., Hendriks, I. E., Díaz-Rúa, R., & Duarte, C. M. (2022). eDNA Reveals the Associated Metazoan Diversity of Mediterranean Seagrass Sediments. Diversity, 14(7), 549.

Stat, M., John, J., DiBattista, J. D., Newman, S. J., Bunce, M., & Harvey, E. S. (2019). Combined use of eDNA metabarcoding and video surveillance for the assessment of fish biodiversity. Conservation Biology, 33(1), 196-205.

Line 91-101: We have added a paragraph on current eDNA seagrass research that expands beyond California, including the citations listed above, for broader context. The paragraph now reads “Environmental DNA has been shown to be a powerful tool when surveying seagrass habitats. Researchers have previously employed this method via water column collection [23-28] and sediment sampling [29-30]. A number of these seagrass studies have demonstrated that when directly compared to a conventional survey method, eDNA was able to detect a higher number of species [23-25]. Other studies emphasized the importance of using concurrent eDNA and conventional survey techniques in revealing the full scope of biodiversity [26, 27, 30]. Despite the literature support of eDNA’s use in seagrass monitoring, there has been no eDNA surveys done on fish communities of Southern California seagrass beds, which sit in a very specific biogeographic position of a productive upwelling region for both island and mainland populations. In Southern California, some of the seagrass population is within marine protected areas and others within heavily human impacted urban coastal environments.”

9. Introduction: The current organization of the introduction requires significant reorganization, starting with a general context to specific research gaps. Additionally, there is a notable absence of relevant literature that should be included to provide a comprehensive understanding of the topic.

Line 91-101: We have reorganized some of the paragraphs in the introduction as well as included a new paragraph on relevant literature to seagrass eDNA listed above.

10. Line 60 – 63: Explain before how is the structural complexity of seagrass meadows (three dimensional habitat – from belowground components with network of roots and rhizomes to aboveground components) and which components are important for fish. Also, not sure what do you mean with unstructured habitats, be more specific. Do you mean bare sand? Other macrophytes?

Lines 57-66 & 388-393: We have added which components of the seagrass are important for fish on lines 57-66 and have updated the unstructured habitats to bare sand. We have also added how structural complexity of seagrass meadows affect fish diversity on lines 388-393. This paragraph reads “Additional influences on fish diversity seagrass sites that may contribute to these differences beyond geographic setting of the meadow includes heterogeneity of environments surrounding the meadow [61], proximity to other seagrass sites [62,63], seagrass canopy height [64], and seagrass cover [65]. Although our in-situ design can’t account for all possible influences on seagrass diversity, our surveys suggest that geographic location can impact up to half the found species at a given seagrass bed.”

11. Line 83-88: The paragraph is better placed in the earlier sections where the significance of seagrass meadows is discussed: they are hotspots for biodiversity but have high regression rates and therefore it is necessary to monitor seagrass meadows continuously due to the changing ocean conditions and their already current regression rates.

Line 66: We have moved the paragraph earlier for better placement starting on line 66.

12. Line 94: edna is not only used to follow biodiversity also to understand the diet of animals, trophic interactions…etc. include citation

Line 75: We have added the citation to the text and how eDNA can be used for other purposes reading “This DNA is extracted and certain gene regions, known as barcodes, are amplified and sequenced to reveal species presence or absence for broad-scale biodiversity, predator diet analysis and trophic interactions [20].”

13. Line 97: include citation

Line 81: We have added the citation to the text which is now on line 81. 

14. Line 104: Also include that visual census surveys are limited in scope as they tend to focus on specific taxa and are conducted at a local scale. Additionally, these surveys require taxonomic experts to accurately identify and classify organisms. Therefore, there is a potential for misidentification, which can introduce errors or uncertainties in the collected data.

Line 81: We have updated to include the limitations of visual surveys of needing taxonomic experts and to misidentify species on line 81 reading “Conventional surveys require taxonomic identification by an expert which could introduce errors from possible misidentification.”

15. Line 106: Include in this section what has been done in edna in seagrass meadows (both in water and sediments) in the world and in California.

Line 91-101: We have added a paragraph on current eDNA seagrass research that expands beyond California, including the citations listed above, for broader context. The paragraph now reads “Environmental DNA has been shown to be a powerful tool when surveying seagrass habitats. Researchers have previously employed this method via water column collection [23-28] and sediment sampling [29-30]. A number of these seagrass studies have demonstrated that when directly compared to a conventional survey method, eDNA was able to detect a higher number of species [23-25]. Other studies emphasized the importance of using concurrent eDNA and conventional survey techniques in revealing the full scope of biodiversity [26, 27, 30]. Despite the literature support of eDNA’s use in seagrass monitoring, there has been no eDNA surveys done on fish communities of Southern California seagrass beds, which sit in a very specific biogeographic position of a productive upwelling region for both island and mainland populations. In Southern California, some of the seagrass population is within marine protected areas and others within heavily human impacted urban coastal environments.”

16. Methods: To understand better the sampling design, Include a table with date of collection, site, location, seagrass meadow/bare sand, open coast/embayment, depth, geographic coordinates, visual survey conducted or not.

Line 145: We have added our sampling design into the methods as Table 1.

17. Line 143-149: What were the depths of the sampling sites? Same depth among sites?

Line 145: We have added the depth of the sampling sites to the methods in Table 1.

18. Line 164: use citation number to be consistent though the text

Line 157: We have added the citation number to the text now on line 157.

19. Line 403: this paragraph fits better at the end of the discussion

We left this paragraph in the same location but have reorganized and added other paragraphs within the discussion.

20. Figure 1: Heading missing

We have added a heading to the Figure 1.

21. Figure 6: how can be that the number of species found in both surveys is bigger than the sum of edna and scuba survey?

The number of species found in both surveys is not bigger than the sum of the eDNA and scuba survey. The number of species found in both surveys are the fish that were jointly found by eDNA and scuba survey. For example, kelp bass and pacific jack mackerel were picked up by both eDNA and scuba surveys at the Amarillo site. eDNA then found an additional 16 fish species that scuba didn’t find while scuba found 2 species that eDNA didn’t. eDNA found 18 fish total while scuba found 4 fish total. 

Line 338: We have edited the text to make this clearer in the caption for figure 6 on line 338. It reads “The number of species observed by eDNA and conventional methods. “Both” indicates species that were detected by both scuba survey and eDNA survey. “Scuba survey only” and “eDNA survey only” show the number of species that were uniquely detected by that method. Total number of species detected by scuba survey is “both” plus “scuba survey only” and total number of species detected by eDNA surveys is “both” plus “eDNA survey only”.

22. Figure 4: Fit fish species on the ordination so that their relative length indicate the correlation between species and the NMDS.

We have added the fish species onto the ordination plot, but only labeled the top five strongest associations on the graph to ensure clarity of the figure.

Reviewer #3: PONE-D-23-14325

This study used eDNA metabarcoding to examine fish species richness in 5 seagrass meadows in Southern California using one genetic marker. The paper is well written, has solid approach for the bioinformatics and statistical analyses. The results are set in the regional literature and provides a nice contribution to the growing literature on the utility of eDNA metabarcoding as a non-invasive method for assessing species richness in sensitive habitats. I have suggestions for revisions to improve the scope and relevance of the study beyond California. Plus some general comments on other environmental characteristics that may be correlated with the differences between embayment and open coastal seagrass meadows that I think should be included in the literature review. No new analyses are necessary, as the sample size is too small (5 meadows) but I think a few key references on how the surrounding seascape and meadow characteristics may influence fish diversity within the sampled seagrass is needed.

I have provided some line edits below:

Thank you for your comments on our paper and its utility in the eDNA literature. We appreciate the revisions you have suggested and have incorporated them into the manuscript.

23. Line 69-81 – Although locally relevant, this paragraph is too detailed for a study introduction. It seems more suited for the discussion where the authors can report their results in the context of other studies. A literature review for this paragraph potentially more relevant would be to focus on how seagrass meadow characteristics and the surrounding seascape may influence biodiversity within the meadow. For example, the heterogeneity of environment outside the meadow, the size of the meadow, shoot density, temperature, salinity (e.g., Proudfoot et al 2023 https://doi.org/10.1007/s12237-023-01203-z, Chalifour et al 2019 https://doi.org/10.3354/meps13064, Gilby et al 2018 DOI: https://doi.org/10.3354/meps12394, and many more)

Line 354-363: We have moved this paragraph to the discussion on lines 354-363. The paragraph reads “Our eDNA survey detected 78 unique species of fish within Southern California seagrass beds- 48 species off the mainland, 48 species off the island and 54 species in the embayment. The number of fish surveyed is on par with or greater than other previous surveys in the area. From 1987 to 2010, embayment seagrass beds in San Diego Bay and Mission Bay were found to have supported 50 species of fish [14]. Newport Bay, the site of our embayment seagrass beds, has been surveyed since 2003; the latest monitoring survey published in 2020 found 26 species of fish [15]. One survey of open coast and island seagrass beds around the Northern Channel Islands and Santa Barbara coastline found that open coast beds supported 20 species of fish while island beds supported 41 species of fish [16]. In 2018, island seagrass beds along Catalina Island were recorded to support 28 species of fish [17,18].”

Line 388-393: We included the other factors influencing biodiversity such as environment outside meadow and size into the discussion. This paragraph reads “Additional influences on fish diversity seagrass sites that may contribute to these differences beyond geographic setting of the meadow includes heterogeneity of environments surrounding the meadow [61], proximity to other seagrass sites [62,63], seagrass canopy height [64], and seagrass cover [65]. Although our in-situ design can’t account for all possible influences on seagrass diversity, our surveys suggest that geographic location can impact up to half the found species at a given seagrass bed.”

24. Line 375 – Some of relevant literature above could also be brought into the interpretation of results and acknowledge limitations of study, as multiple unmeasured factors likely influence observed species differences between sites.

Line 388-393: We have added limitations of the study by unmeasured factors into the discussion reading This paragraph reads “Additional influences on fish diversity seagrass sites that may contribute to these differences beyond geographic setting of the meadow includes heterogeneity of environments surrounding the meadow [61], proximity to other seagrass sites [62,63], seagrass canopy height [64], and seagrass cover [65]. Although our in-situ design can’t account for all possible influences on seagrass diversity, our surveys suggest that geographic location can impact up to half the found species at a given seagrass bed.”

25. Line 524 – the authors refer to eDNA metabarcoding as a useful a biodiversity monitoring tool for seagrass meadows, but in this paragraph on limitations, they fail to mention that eDNA metabarcoding is currently more of an assessment tool, or a snapshot of species richness. It can be examined over time but species richness is limited in its value for understanding ecosystem function. Just an acknowledgement that eDNA metabarcoding for biodiversity monitoring currently limited to richness (and at most rank abundance of dominant species – Skelton et al 2022 https://doi.org/10.1002/edn3.355

Line 557-562: We have added this as a limitation to eDNA surveys. This reads “A third limitation of eDNA is that, in its current state, it is a snapshot of species richness which limits our understanding of the data and its ecosystem function that may otherwise be understood from additional data taken from conventional surveys including size frequency, sex ratio, and absolute abundance data. Despite this, the information from eDNA still provides a valuable insight into local biodiversity.”

26. Figure 1 – inset map needed. Readers outside of Southern California need to be oriented to study site. Scale also needed (distance between sites could also be reported in results)

We have added an inset map and scale bars to the maps of Figure 1.

27. Not sure why figure captions are scattered throughout document rather at the end just before the figures. This made it difficult to find the correct caption.

PLOS one submission guidelines state “Place figure captions in the manuscript text in read order, immediately following the paragraph where the figure is first cited. Do not include captions as part of the figure files or submit them in a separate document.”

Figure 2 – Include “Violin plot” in figure caption to ensure caption is descriptive of plot.

Line 284 & 288: We have added violin plot to the figure caption.

---

## [Editor Report · Decision Letter 1]

18 Sep 2023

Environmental DNA metabarcoding reveals distinct fish assemblages supported by seagrass (Zostera marina and Zostera pacifica) beds in different geographic settings in Southern California

PONE-D-23-14325R1

Dear Dr. Waters,

We’re pleased to inform you that your manuscript has been judged scientifically suitable for publication and will be formally accepted for publication once it meets all outstanding technical requirements.

Kind regards,

Silvia Mazzuca

Academic Editor

PLOS ONE
---

## [Editor Report · Acceptance letter]

25 Sep 2023

PONE-D-23-14325R1 

Environmental DNA metabarcoding reveals distinct fish assemblages supported by seagrass (*Zostera marina* and *Zostera pacifica*) beds in different geographic settings in Southern California 

Dear Dr. Waters:

I'm pleased to inform you that your manuscript has been deemed suitable for publication in PLOS ONE. Congratulations! Your manuscript is now with our production department. 

Kind regards, 

on behalf of

Prof Silvia Mazzuca 

Academic Editor

PLOS ONE